# Educational Processes for Health and Disease Self-Management in Public Health: A Systematic Review

**DOI:** 10.3390/ijerph18126448

**Published:** 2021-06-15

**Authors:** Jessica Alejandra Ruiz-Ramírez, Yury Arenis Olarte-Arias, Leonardo David Glasserman-Morales

**Affiliations:** 1School of Humanities and Education, Tecnologico de Monterrey, Monterrey 64849, Mexico; glasserman@tec.mx; 2Research Department, Escuela Colombiana de Rehabilitación, Bogota 111321, Colombia; yury.olarte@ecr.edu.co

**Keywords:** self-management of health, educational processes, self-management of diseases, innovative education, higher education

## Abstract

This study systematically reviewed processes and educational programs for self-management of health and diseases that are the subject of public health attention. This systematic review of the literature (SRL) is relevant to recognizing the characteristics of the educational processes in self-managing chronic diseases in contexts where technology did not play a significant role. Following the PRISMA protocol, the authors independently reviewed full-text articles from several databases using the following criteria: (1) intervention studies evaluating the effects of self-management health programs; (2) educational process of disease self-management; (3) studies that included at least one control group, and (4) peer-reviewed studies. In addition, the Scottish Intercollegiate Guideline Network measurement tool was used to assess the risk of bias in each trial. In the final sample, 38 articles were included. The findings regarding health education methods of self-care, using community-based care and technological tools, are considered fundamental. Among the conclusions, the relevance of the pedagogy that health education processes demand improvement in post-pandemic program effectiveness stands out.

## 1. Introduction

Educational processes in public health seek to ensure the populations’ right to health by promoting improved living conditions, welfare, and development [1]. Public health education programs address healthcare-related challenges caused by (often chronic) diseases; and they focus on socially and economically vulnerable populations [2]. They also promote healthy lifestyles through nutrition and mental, sexual, and reproductive health. Similarly, medical and health education processes are mainly organized around disease prevention and health promotion training programs. Their common purpose is to facilitate knowledge generation and develop skills necessary for the self-care and self-management of diseases [3]. Likewise, training programs are focused, on the one hand, on health education programs, i.e., actions and activities focused on providing information to individuals and groups to encourage the promotion, maintenance, and restoration of health, and, on the other hand, on medical education programs, which aim to promote the training of physicians and other health professionals for participation in health education contexts, including those required in public healthcare.

Health education interventions have traditionally focused on prescriptions and lifestyle restrictions [4]. This approach is a passive transfer of theoretical knowledge, but lacks feedback and does not always consider the realities of communities [5]. Therefore, health authorities have had to make great efforts to facilitate access to medical and health education, considering that educational processes optimize medical treatment results and the prevention of the development of diseases or problems that affect people’s quality of life [6]. Such actions align with the World Health Organization (WHO) recommendations, which state that “quality of life refers to a person’s perception of his or her life situation concerning the culture and value system of society, as well as its relationship to goals, expectations, standards and needs” ([7], p. 3).

Therefore, educational actions developed in public health contexts and reported by research mainly focus on disease self-management and the appropriation of self-care skills [8]. Medical and health education literature primarily verify the numerous challenges people with diseases or public health problems face in daily self-care activities. This suggests the need for ongoing support to navigate the self-care process, seeking to manage the personal, environmental, and economic factors that challenge disease management in everyday contexts [9]. Therefore, experts suggest that health care should include routine medical treatment and health education to maintain comprehensive wellness [10]. Thus, the educational processes should include addressing the challenges that affect the quality of life and motivation to develop self-care activities and controlling the physical symptoms of disease [11].

Access to medical and health education programs include mechanisms to counteract their low participation rates. According to health institutions and research worldwide, low community participation is mainly related to socioeconomic conditions, transition effects over time, and a low level of health experts in educational training [12]. Additionally, some barriers to health education program effectiveness were accentuated and diversified due to health conditions caused by the COVID-19 pandemic. Health conditions that, in turn, have made visible and deepened economic, social, and cultural factors associated with inequality, limiting access to social and health services, and health education.

The importance of working on the aspects that affect people’s willingness to participate in educational programs and daily, personal, collective activities to control and treat disease is evident. In this regard, improving self-regulation and self-management requires immediate feedback to modify behavior [13]. Thus, the concept of self-management emerges in health systems in the migration of the orientation from treatment to health promotion and maintenance [14].

Likewise, health education programs have various innovative educational strategies that focus on the subject and his/her context to facilitate the formative effectiveness of the programs [15]. Education to preserve and promote health is the initial approach implemented at generalized and diverse societal levels to change habits and routines [16]. Subsequently, different community participatory approaches have been incorporated into medical and health education processes. These allow people to have a voice to elucidate barriers, facilitating self-management of health and health information needs, and thus encouraging communities to take on active roles in the educational process, even supporting the design of activities [17].

Other health education initiatives have reflected on the role of health education professionals to ensure that there are no gaps in access to educational programs [18]. Furthermore, it is considered that the training processes of health professionals should be continuous and cover not only the clinical or physical aspects, but also the psychosocial aspects of a population [19]. In this regard, health education should be provided holistically by various professionals who have both updated knowledge and adequate skills. Further, health education programs have involved the use of technologies to increase the effectiveness of these programs. Information and communication technologies are powerful tools for promoting the quality and efficiency of education. They have helped migrate knowledge in education to transform traditional educational methods used in the health field [20]. Thus, web-based applications and technologies are making inroads as health education tools for self-management, self-care, and control of diseases to improve the quality of life of individuals. In addition, ICTs provide valuable tools for accessing information and improving the appropriation of health-related knowledge [21].

Educational approaches involving technological tools in training processes in healthcare are in an exploratory phase, given the possibilities of improvement offered by technology [22]. This systematic review of the literature is relevant to recognizing the characteristics of the educational processes in self-managing chronic diseases in contexts where technology did not play a significant role. This research will contribute to educational methodologies in public health for a new vision of implementing educational technologies in a post-pandemic context. Therefore, this article helps expand the theoretical corpus on the characteristics of innovative educational processes to self-manage health in public healthcare. The first section of this article describes the methodological strategy used in the research based on a systematic literature review, while the second presents our findings, analysis, and interpretation. Finally, the third section contains discussions on the following question: what are the characteristics of the educational processes of self-management of health in public health care?

## 2. Materials and Methods

### 2.1. Literature Search

This systematic review was conducted according to the preferred reporting items for systematic reviews and meta-analyses, or PRISMA, guidelines [23]. To identify potentially relevant papers, we searched the following bibliographic databases: Scopus and Web of Science (WoS). Articles from peer-reviewed journals were included if they were published between 2006 and 2020, written in English, and available via open access to use information and communication technologies (ICTs) to increase and improve the dissemination of knowledge. Open access refers to freedom, flexibility, and impartiality [24]. Furthermore, quantitative, qualitative, and mixed methods studies were included to consider different aspects of self-management health education programs. Finally, articles that did not fit the study’s conceptual framework were excluded, for example, research developed within a pandemic context.

### 2.2. Selection Criteria

One author performed the initial eligibility assessment by reviewing the titles and abstracts. Then, two authors independently reviewed full-text versions of 92 articles using the following criteria: (1) intervention studies evaluating the effects of self-management health programs; (2) studies including at least one control group; and (3) peer-reviewed studies.

### 2.3. Data Extraction

Data extraction was performed independently by the authors. The following data were extracted from each study: first author, date and place of publication, study design, population served, analysis of the program effectiveness, EdTech used, and pedagogical strategy. Extracted data were entered into standardized MS Excel files. Any disagreements were resolved by discussion among the authors.

### 2.4. Quality Assessment Tool

The Scottish Intercollegiate Guideline Network (SIGN) measurement tool was used to assess the risk of bias in each study in this review. SIGN was developed in 1993 to improve healthcare quality for populations in Scotland by reducing variation in practice and outcomes by developing and disseminating national clinical guidelines with recommendations for effective practice based on current evidence [25]. This tool suggests the use of risk of bias domains, and so, a set of key considerations was used. These considerations sought to prompt and guide assessors to consider potentially relevant issues when conducting their assessment for a risk of bias domain. Using SIGN, we assessed the study’s internal validity and risk of bias. We assigned values of “high quality (++)”, “acceptable (+)”, “low quality (−)”, or “unacceptable: rejection (0)” to each study. The rating (0) enabled assessors to indicate that the information required to rate risk of bias was either not reported at all, or reported too incompletely to rate risk of bias with sufficient confidence. The risk of bias was assessed independently by the reviewers and a consensus process resolved any disagreement.

## 3. Results

### 3.1. Identification and Selection of Studies

A total of 3114 records were found in the database search. These search results were imported using Parsifal, and 1428 duplicates and 81 records marked as ineligible by automation tools were removed. A detailed flow chart of the selection process is shown in Figure 1.

After the elimination of duplicates, 1605 records were examined. Subsequently, we searched for papers citing any of the initially included studies. However, no additional articles meeting the inclusion criteria were found in these searches. Finally, 38 full-text papers were reviewed.

### 3.2. Characteristics of the Study

It was identified that, while some articles met the inclusion criteria, they evaluated performances in regular professional training processes in health [26,27], leaving aside the analysis of teaching-learning processes in public health contexts. For this reason, they were excluded from the analysis. The research was conducted across five continents: Europe (*n* = 14), Asia (*n* = 13), North America (*n* = 8), Africa (*n* = 3), and Oceania (1). Twenty-five studies involved adults aged between 35 and 60 years, while ten studies reported educational programs for self-management of health and disease with young adult participants between 18 and 35. In a smaller proportion, programs involving follow-up and control of people >65 years (*n* = 2) and accompanied by health professionals (*n* = 2) were identified. The general characteristics of the studies are presented in Table 1.

Table 2 contains a summary of the research included in this review, including information on the variables: author, year, country of study development, study design, population served, analysis of program effectiveness, tools, technologies used, and pedagogical strategy. The selected studies were published between 2016 and 2020.

#### 3.2.1. Pedagogical Strategies for Disease Self-Management Programs

Disease self-management programs link different pedagogical strategies and learning methodologies developed to contribute to the knowledge and control of chronic diseases. It was highlighted that 16% of the investigations [7,31,33,34,47,48] related to using the DSME (diabetes self-management education) methodology to self-manage Type 1 and Type 2 Diabetes, focusing on glycemic control measures because of changes in glycosylated hemoglobin. Likewise, the COPERS (Coping with persistent Pain, Effectiveness, Research into Self-management) methodology [32] and the TPE (Therapeutic Patient Education) methodology [39] were used to identify the needs of patients and family members in pain management. Concerning the above, the results showed that only 21% of the research on medical and health education [9,15,20,22,30,47,48,56] described educational programs or actions oriented around community-based health and appropriation of culture [6].

Similarly, methodologies that allowed discussion and knowledge exchange among patients in collaborative groups in contextual intervention programs were recognized. We also found analyses of patient perception through the accompaniment of a professional or the majoritarian use of qualitative analysis tools. Figure 2 presents the trends in methodological design.

#### 3.2.2. EdTech to Support Educational Processes for Self-Management of Health or Self-Management of Illnesses

This review showed that 29% of the texts studied [20,22,28,29,34,37,40,47,54,55,57] reported technological tools as a digital literacy alternative for people with health conditions, as well as for professionals, family members, and caregivers. Technological tools accessible to the community were identified using free applications, such as WhatsApp and WeChat, accessible via smartphone, and disseminating content through websites and social networks, defined as Digital Health Interventions (DHI).

It was emphasized that the analysis of technological tools, which contributed to self-management of diseases, were contemplated where technology was not an indispensable tool for their management, as reflected in 71% of the investigations that do not report the mediation of technology in their educational process. This point becomes relevant for identifying lessons learned and best practices of current educational processes that aim to meet educational challenges during and after pandemics, where technology is the main instrument for transmitting knowledge.

### 3.3. Quality Assessment

According to the SIGN checklist, 18 of the studies [7,20,28,30,31,33,35,37,38,39,40,41,42,43,44,45,49,55] were rated “high quality” and 17 studies were “acceptable quality” [5,6,15,19,29,34,36,46,47,48,50,51,52,53,54,56]. Although the studies were rated as being high and acceptable quality, for sample selection random assignment was used and no detailed description of the procedure was provided. Three of the studies were rated as low quality [9,22,32]. The main reasons for the “low quality” ratings were inadequate randomization or the method of concealment used. Among the 11 categories of the SIGN checklist, the studies were classified as “high risk of bias”, particularly in two categories: “when the study is conducted at more than one site, are the results comparable for all sites?” and “how well was the study conducted to minimize bias?”. These categories were identified following the results assessment by each author. Figure 3 presents the risk of bias graph.

## 4. Discussion

In interpretating the findings, we were careful about generalization due to the number of studies reviewed and their open-access character. However, the results allowed us to answer the guiding research question and identify the educational process characteristics of self-management of health in public healthcare. The results aligned with recent research on the innovation needs of medical and health education programs reported by health institutions and agencies worldwide [28,30]. The articles reviewed describe educational programs focused on self-management and self-control to contain disease, especially chronic disease care. These articles suggest that public health education processes continue to prioritize remedial objectives over treatment outcomes for health maintenance.

While public health processes aim to prevent diseases, promote and maintain health and quality of life, the above is interpreted as a call to public health and medical and health education to diversify their objectives and educational strategies for the prevention, promotion, maintenance, and monitoring of population health. Doing so aligns with the recommendations of the World Health Organization (WHO). It invites broadening of the clinical and therapeutic perspective of healthcare and health education through strategies that build frameworks of trust, social support, and learning which respond to the needs of individuals and communities for health and welfare beyond the health-illness binomial [27].

Thus, it seems that health education programs continue to emphasize informative actions on the self-management of diseases, leaving at a low priority the training and monitoring of learning and the appropriation of knowledge and skills for self-care and self-management of health in the daily activities of families and communities [22]. Some recent research initiatives have sought to decenter the functionalist perspective of health education programs. However, little research has helped guide the development and implementation of practical health promotion efforts [23]. Most of the findings report educational programs focused on measuring changes in signs and symptoms of disease and, to a lesser extent, assessing behavioral changes to achieve self-management of health. There are few learning assessment products developed in health training.

The findings of this review indicate that health education programs in public health face significant challenges in guaranteeing program access, participation, and permanence and improving the effectiveness of the learning processes. The programs studied mainly address objectives for the self-management of diseases, remaining at the level of health maintenance, and lacking educational processes for disease prevention, promoting integral health, and monitoring people’s quality of life and welfare. Some studies emphasized the importance of involving the participants in the design of the educational sessions and using tools, such as dialogue and other interactive methods, to improve participation [28]. Thus, methods, such as peer education, are proposed to strengthen the emotional bonds that alleviate the psychological pressure caused by the health affectation [38]. Other proposals include training and solid supervision of health educators to develop contextual patient-centered approaches. However, the results showed that less than a quarter of medical and health education research describe programs or educational actions oriented to community-based health.

Regarding technological tools, electronic education has become crucial in the information era and, therefore, it is also incorporated in medical and health education processes. Our results identify that technologies have offered robust mechanisms that support the self-management of health and guarantee a lasting educational experience for people, any time and place [30], with new ways to favor self-management processes in public health. Furthermore, tools such as the Internet and technological devices have made it possible to easily and quickly view or generate content (text, images, voice, and video) to support health literacy and self-care, and contribute to the population’s quality of life. However, these technologies require digital literacy processes for people with health conditions, professionals, family members, and caregivers.

The findings report that educational processes for self-management of health hold promise for public health intervention when technology supports it. This review suggests recognizing and integrating the transformative possibilities of community education methods coupled with the use of technology. In addition, it is relevant to consider the pedagogical character of health education processes to improve the effectiveness of the programs. To this end, it is essential to strengthen the educational training processes for health professionals and structure evaluation goals for the teaching-learning process in these programs. The assessments should cover content, teaching strategies, appropriation of knowledge, and self-management of health in populations. Concerning technologies in the training processes, their functionality and usability, with regard to achieving the expected learning, should be evaluated.

The findings regarding health education methods of self-care as part of community-based care and how technological tools are used are fundamental to this review. Consequently, we urge future research to focus on studying the barriers and facilitators that arise when using technological tools as mediators of learning in health education programs, mainly in terms of access to infrastructure, generational gaps, cultural factors, levels of digital literacy, and economic disadvantages [23].

One limitation of this systematic review is language bias, as we only included studies published in English. Therefore, studies published in other languages, such as Chinese and Japanese, were not included in this review. In addition, the review did not include research from grey literature, which could exclude important unpublished pieces produced by non-governmental organizations. Finally, it should be noted that the analyses performed represent a partial view of the phenomenon under study and, therefore, the generalizability of the data is not considered. Despite these limitations, this literature review study increased the understanding of self-management health education processes and identified opportunities for improvement.

## 5. Conclusions

This review began with the question, “what are the characteristics of educational processes of self-management of health in public health care?”. We identified that the most relevant characteristics respond to objectives focused on self-management of diseases, primarily chronic, and mitigating disease signs and symptoms. There is a focus on remedial education for the maintenance of health that is typical of hospital environments, which, in the public health context, is limited. In this regard, institutions and researchers propose innovative medical and health education processes to improve training and contribute to the prevention, promotion, care, maintenance, and monitoring of integral public health.

The innovative processes highlight social and community-based learning characteristics that involve peers, caregivers, families, and professionals in training actions. In addition, some initiatives highlight socio-emotional characteristics of education and the need for contextual learning in daily life practices. Finally, there are characteristics oriented around using technological tools in the teaching-learning process as an area of opportunity to improve their effectiveness and increase their coverage.

Future research is invited to examine the barriers and facilitators in medical and health education programs using socio-community and technology-enhanced educational strategies. Furthermore, the contributions of innovative processes for public health education toward long-term improvements in self-management of health and diseases should be evaluated. These research needs are especially relevant due to the COVID-19 health crisis because the fundamental medium of knowledge transmission for the formulation and implementation of health education programs, which recognize the health, economic, social, and cultural realities of communities, has become technology.

## Figures and Tables

**Figure 1 ijerph-18-06448-f001:**
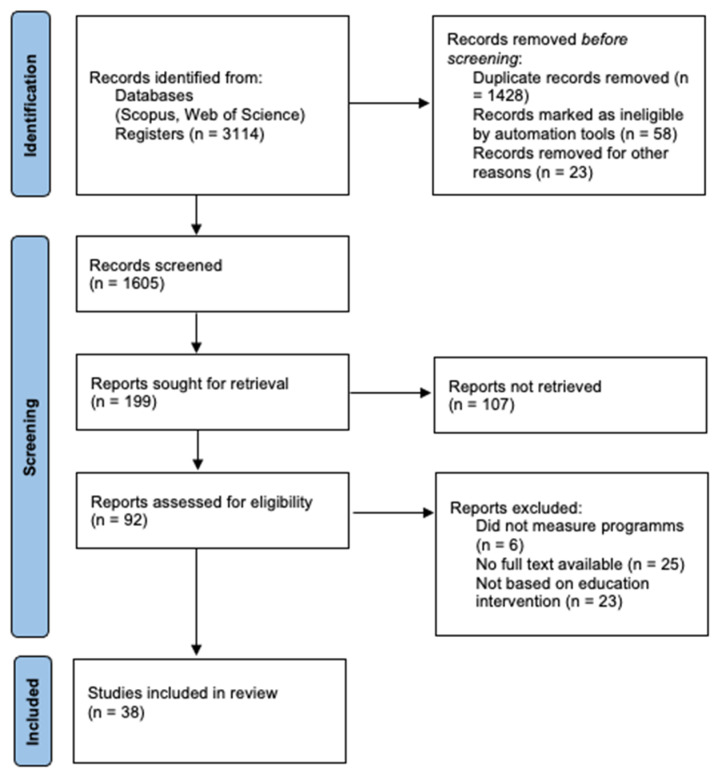
Flow diagram of the screening process for systematic reviews and meta-analyses (PRISMA).

**Figure 2 ijerph-18-06448-f002:**
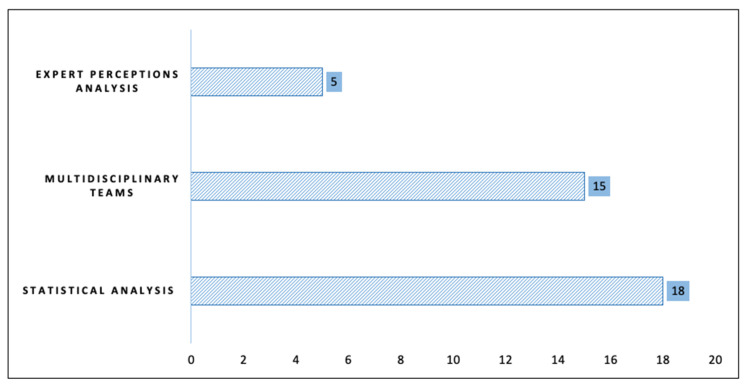
Methodological design.

**Figure 3 ijerph-18-06448-f003:**
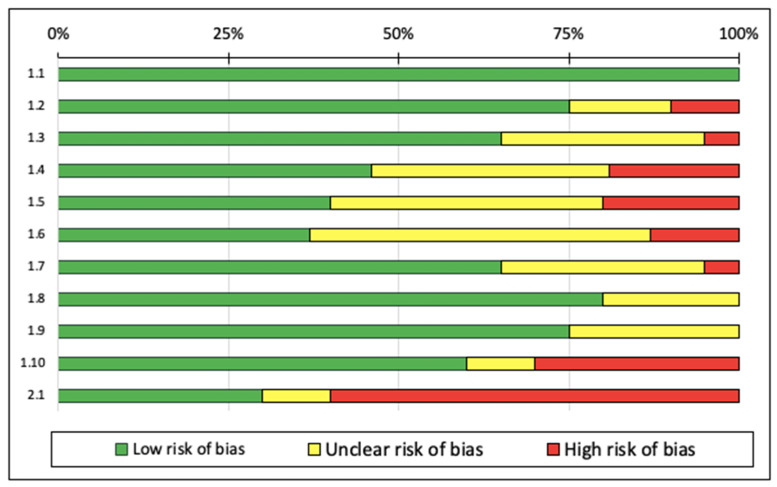
Risk of bias graph. Note: The authors’ judgments on each element of risk of bias are presented as percentages for all included studies. (1.1) the study addresses an adequate and focused question; (1.2) the allocation of participants in the treatment group is randomized; (1.3) an adequate method of concealment is used; (1.4) the design keeps participants and investigators “blind” about treatment allocation; (1.5) the treatment and control groups are similar at the start of the trial; (1.6) the only difference between groups is the investigational treatment; (1.7) all relevant outcomes are measured in a standard, valid, and reliable manner; (1.8) what percentage of individuals or groups enrolled in each treatment arm of the study dropped out before the study completion? (<20% low risk of bias); (1.9) all participants are analyzed in the randomized groups (often referred to as intention-to-treat analysis); (1.10) when the study is conducted at more than one site, the results are comparable for all sites; and (2.1) how well was the study conducted to minimize bias?

**Table 1 ijerph-18-06448-t001:** General characteristics of the included studies (*n* = 38).

Variables	Categories	*n*	%
Evaluation of the self-management program	Statistical analysis	18	(47.3)
	Multidisciplinary teams	15	(39.4)
	Perception analysis	3	(7.8)
	Expert analysis	2	(5.2)
Study design	Controlled trial	20	(52.6)
	Qualitative	9	(23.6)
	Observational study	5	(13.1)
	Community clinic	1	(2.6)
	Study of parallel groups	1	(2.6)
	Mixed type research	1	(2.6)
	Grounded theory	1	(2.6)
Diseases	Diabetes mellitus type 2	12	(31.6)
	Chronic diseases	9	(23.7)
	Type 1 diabetes	7	(18.4)
	Cancer	2	(5.3)
	Diabetes and acute coronary syndrome	1	(2.6)
	Epilepsy	1	(2.6)
	Maternal Stress	1	(2.6)
	Hypertension	1	(2.6)
	Obesity	1	(2.6)
	Osteoarthritis	1	(2.6)
	Metabolic syndrome (MetS)	1	(2.6)
	HIV and tuberculosis	1	(2.6)

**Table 2 ijerph-18-06448-t002:** Summary of included research.

Author, Year	Country of Study Development	Study Design	Population Served	Analysis of the Program Effectiveness	EdTech Used	Pedagogical Strategy
Woods (2019) [28]	South Africa	Study observational	Health professionals	Statistical analysis	WhatsApp	Exchange and discussion of critical clinical cases through a social networking platform
Zimbudzi (2019) [29]	Australia	Theory	Adult patients	Equipment multidisciplinary	DVD, websites, telephone support	Design-based research (DBR) to develop an educational resource
Risica (2018) [30]	United States	Community clinic	Hispanic adults	Statistical analysis	Without mention	Intervention program called Vida Sana Program
McElfish (2019) [31]	United States	Controlled trial	Adult patients	Statistical analysis	Without mention	DSME (diabetes self-management education) methodology
Patel (2019) [32]	England	Controlled trial	Adult patients	Perception analysis	Without mention	COPERS methodology (Coping with Persistent Pain, Effectiveness, Research into Self-management).
Hailu (2019) [33]	Ethiopia	Controlled trial	Adult patients	Statistical analysis	Without mention	DSME (diabetes self-management education) methodology
Pal (2018) [34]	England	Qualitative	Adult patients	Equipment multidisciplinary	Digital health interventions (DHI)	DSME (diabetes self-management education) methodology
Azami (2018) [35]	Iran	Controlled trial	Adult patients	Statistical analysis	Without mention	Professional accompaniment
Lee (2019) [36]	Korea	Controlled trial	Adult patients	Equipment multidisciplinary	Without mention	Professional accompaniment
Omidi (2020) [37]	Iran	Controlled trial	Adults—women	Statistical analysis	Social networks	Technology-mediated professional accompaniment
Ji (2019) [7]	China	Controlled trial	Adult patients	Statistical analysis	Without mention	DSME (diabetes self-management education) methodology
Farahmand (2019) [38]	Iran	Controlled trial	Senior citizens	Statistical analysis	Without mention	Professional accompaniment
Prevost (2019) [39]	France	Controlled trial	Adult patients	Statistical analysis	Without mention	TPE (Therapeutic Patient Education) methodology
Boels (2018) [40]	Netherlands	Controlled trial	Adult patients	Statistical analysis	Smartphone app	Monitoring mediated by technological tools
Zheng (2019) [41]	China	Controlled trial	Adult patients	Statistical analysis	Without mention	Professional accompaniment
Jinnouchi (2019) [42]	Japan	Study of parallel groups	Adult patients	Statistical analysis	Without mention	Use of educational materials: books and DVDs
Smith (2018) [43]	United States	Controlled trial	Adults—men	Statistical analysis	Without mention	Chronic Disease Self-Management Education (CDSME) program
Kennedy (2019) [44]	United States	Controlled trial	Adults—women	Statistical analysis	Without mention	Participation in collaborative groups
Pozza (2020) [45]	Italy	Controlled trial	Adult patients	Statistical analysis	Without mention	Analysis using questionnaire results
Sanders (2018) [46]	England	Qualitative	Young adults	Equipment multidisciplinary	Without mention	Educational intervention: ‘WICKED’ (Working with Insulin, Carbohydrates, Ketones and Exercise to Control Diabetes)
Ross (2019) [47]	England	Qualitative	Adult patients	Equipment multidisciplinary	Websites	DSME (diabetes self-management education) methodology
Gagliardino (2019) [6]	Middle East	Study Observational	Adult patients	Expert analysis	Without mention	Analysis using questionnaire results
Senteio (2018) [22]	United States	Research mixed type	Senior citizens	Perception analysis	Smartphone app	Community-based health education program
Carmienke (2020) [48]	Germany	Qualitative	Adult patients	Equipment Multidisciplinary	Without mention	DSME (diabetes self-management education) methodology
Pratt (2018) [49]	United States	Controlled trial	Adult patients	Statistical analysis	Without mention	Professional accompaniment
Siltanen (2020) [19]	Finland	Qualitative	Health professionals	Expert analysis	Without mention	Professional accompaniment
Muchiri (2019) [9]	South Africa	Qualitative	Adult patients	Perception analysis	Without mention	Analysis using questionnaire results
Varming (2018) [15]	Denmark	Qualitative	Educators of health programs and adult patients	Equipment multidisciplinary	Without mention	Participation in collaborative groups
Jönsson (2019) [50]	Sweden	Study Observational	Adult patients	Equipment multidisciplinary	Without mention	Participation in collaborative groups
Rasoul (2019) [20]	Iran	Qualitative	Adult patients	Statistical analysis	Internet webpage and social networks	Monitoring mediated by technological tools
Ridsdale (2018) [51]	United Kingdom	Controlled trial	Adults and caregivers	Equipment multidisciplinary	Without mention	Participation in collaborative groups
Chang (2019) [52]	Taiwan	Controlled trial	Adult patients	Equipment multidisciplinary	Without mention	Inverted classroom
Davies (2019) [53]	United Kingdom	Controlled trial	Adult patients	Equipment multidisciplinary	Without mention	Professional accompaniment
Vandenbosch (2018) [54]	Northern Europe	Study Observational	Adult patients	Equipment multidisciplinary	Websites	Technology-mediated professional accompaniment
Liu (2018) [55]	China	Study Observational	Adult patients	Statistical analysis	WeChat app	Technology-mediated professional accompaniment
Preechasuk (2019) [56]	Thailand	Qualitative	Health professionals	Equipment multidisciplinary	Without mention	Analysis using questionnaire results
Bourbeau (2018) [57]	Canada	Controlled trial	Adult patients	Equipment multidisciplinary	Websites	Professional accompaniment
Zhao (2019) [5]	China	Controlled trial	Adult patients	Equipment multidisciplinary	Without mention	Participation in collaborative groups

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
