# Peer review of "Educational Processes for Health and Disease Self-Management in Public Health: A Systematic Review"

_ijerph, 2021, doi:10.3390/ijerph18126448_

Round 1
Reviewer 1 Report
This is a very well-written article. I have a few minor comments.
1) The authors talk about the role of COVID-19 on the educational programs. I believe its impact is beyond related health conditions. Can you briefly mention its economic, social, and cultural consequences?
2) The authors seem to use medical and health education programs interchangeably. Is it accurate, considering the paper is situated within the context of public health?
3) Did you exclude gray literature? I believe there are important unpublished pieces produced by non-governmental organization, especially given the topic.
4) Can you discuss the limitations more carefully? How about publication bias? Are these results generalizable?
Author Response
File attached

Reviewer 2 Report
This paper aimed to review the processes and educational programs for self-management of
health and diseases that are the subjects of public health attention. The authors followed the
PRISMA PROTOCOL and independently reviewed full-text articles from Scopus and Web of
Science databases. Four criteria were used. Authors used the Scottish Intercollegiate Guideline
Network measurement tool to assess the risk of bias in each trial. The paper is written in a clear
way and is methodologically correct. The authors identified that the most relevant characteristics
respond to objectives focused on self-management of diseases, primarily chronic, and mitigating
signs and symptoms. The findings regarding health education methods of self-care using
community-based care and technological tools are considered fundamental. It was also found out
that the relevance of the pedagogy that health education processes demand to improve post-
pandemic program effectiveness stands out.
Broad comments:
It is a well written paper. The subject taken is important nowadays. The main strengths of the
paper are: i) clear problem statement, ii) very good literature study and review, iii)
methodological correctness, iv) deep discussion of results, v) conclusions emphasizing
contribution of this paper and directions of future research vi) fluent English. Therefore I
recommend paper acceptance in present form.
Specific comments:
Line 207: “This review shows that 29% of the texts studied [20,22,28,29,34,37,40,47,54,55,57)” -
Please change the parentheses to square brackets.
Lines 137-145 – The paper could be supplemented with a more detailed explanation of how the
SIGN measurement tool was implemented in the study. It would support a better understanding
of the paper for those readers who are not familiar with this tool.
Lines 232-231 - The paper could be supplemented with a more detailed explanation about the
results of quality assessment of individual studies, which were included in the research. It would
be also advised to expand on how the evaluation of studies in each criterion was carried out.
Lines 224-225- This statement could be clarified: “Although the studies used random assignment,
no detailed description of the procedure was provided”.
Author Response
File attached.
